# Does It Matter Who You Provide Care for? Mental Health and Life Satisfaction in Young Adult Carers Associated with Type of Relationship and Illness Category—A National Student Survey

**DOI:** 10.3390/ijerph20053925

**Published:** 2023-02-22

**Authors:** Bente Storm Mowatt Haugland, Mari Hysing, Børge Sivertsen

**Affiliations:** 1Department of Clinical Psychology, Faculty of Psychology, University of Bergen, Årstadveien 17, 5009 Bergen, Norway; 2Department of Psychosocial Science, Faculty of Psychology, University of Bergen, 5020 Bergen, Norway; 3Department of Health Promotion, Norwegian Institute of Public Health, 5015 Bergen, Norway; 4Department of Research & Innovation, Helse Fonna HF, 5504 Haugesund, Norway

**Keywords:** young adult carers, mental health problems, life satisfaction, care-receiver, type of illness, type of relationship, caregiving context, national student survey

## Abstract

There is limited knowledge on how caring contexts impact young adults providing informal care for persons with chronic conditions. This study examines associations between outcomes in young adult carers (YACs) and type of relationship (e.g., close or distant family member, partner, or someone outside the family) and type of illness in the care-receiver (e.g., mental, physical illness/disability, or substance abuse). A total of 37,731 students (age 18–25, mean 22.3 years, 68% females) in higher education in Norway completed a national survey on care responsibilities, hours of daily caring, relationship and type of illness, mental health problems (Hopkins Symptoms Checklist-25) and life satisfaction (Satisfaction With Life Scale). More mental health problems and lower life satisfaction were found among YACs compared to students without care responsibilities. The poorest outcomes were reported by YACs caring for a partner, followed by YACs caring for a close relative. Hours spent on daily caring was highest when caring for a partner. Poorer outcomes were reported by YACs caring for someone affected by substance abuse, followed by mental health problems and physical illness/disability. At-risk groups among YACs should be acknowledged and offered support. Future studies are needed to investigate the potential mechanism for the associations between care context variables and YAC outcomes.

## 1. Introduction

A growing body of research has focused on the experiences, needs, and health outcomes of young adult carers (YACs) [1,2,3,4]. These are young adults usually defined as youths between 18 and 25 years old who provide regular informal care, assistance, or support to persons with chronical illness, including mental and physical illness, disability, or alcohol or drug abuse [1,5]. Although most studies have examined YACs who care for a close family member, usually a parent or a grandparent, the person receiving care may also be a sibling, a partner, another relative, a friend, or a neighbor [1]. However, whether the care-receiver is a close relative, a more distant relative, or someone outside the family will probably influence the caregiving context and perhaps also the adjustment of YACs.

Different categories of chronic illnesses are usually combined in research studies on YACs. However, they may potentially represent distinct challenges for the carers. Caring for someone with mental health problems and/or substance abuse may for example involve stigma, shame, and increased focus on emotional care, whereas caring for someone with physical illness may require more personal care tasks. In general, the types of caring activities provided by YACs vary from household chores and activities of daily living (e.g., cleaning and cooking) to intimate and personal care (e.g., feeding, bathing, and dressing), emotional support (e.g., comforting and supervising), and administrative tasks (e.g., paying bills) [1,3,5]. Both positive and negative health outcomes in carers younger than 18 years have previously been associated with type of caring activities reported by the youth [6].

Recent systematic literature reviews on YACs strongly suggest that providing informal care for a person with chronic illness impact YACs negatively, with increased risk for a range of mental health problems and concerns, e.g., worrying, stress, anxiety, depression, anger, resentment, loneliness, and resignation, as well as somatic health complaints, e.g., fatigue, exhaustion, and backaches [1,2,3,4]. A previous study on Norwegian college and university students (*N* = 40,205, age 18–25 years) found an increased risk for mental health problems, insomnia, and somatic complaints, as well as lower life satisfaction among YACs compared to peers without care responsibilities [7].

In addition to the evidence indicating a negative impact on YAC adjustment and health, positive consequences of being a YAC have also been emphasized. Some of these include feelings of satisfaction, emotional maturation, personal growth, development of positive coping strategies and empathy, and learning important life skills, as well as developing a close and meaningful relationship with the care-receiver [3,4,5]. The diversity in outcomes suggests a need for more research to identify high-risk groups, as well as predictors of health outcomes and well-being among YACs.

Young adulthood represents an important developmental phase, with crucial changes and life choices, e.g., choices regarding education, career, romantic relationships, and parenthood. For young people in Western countries, this may be a period of separation and individuation, characterized by increased economic, practical, psychological, and emotional independence, with young adults aged 18 to 25 years commonly referred to as emerging adults [8]. For some, the duties and responsibilities of being a YAC may disturb or postpone crucial developmental tasks during the phase of emerging adulthood [1,5]. Accordingly, YACs report having poorer study progression, more failed exams, more feelings of loneliness, less participation in social and recreational activities, and less time for friends, relaxation, and rest compared to young adults without care responsibilities [5,9].

The need to examine variables related to the care-receiver and the caregiving context has been emphasized [2,10]. Among others, this includes the type of relationship between the YAC and the care-receiver and the type of illness in the person receiving the care. These are variables that may be of importance to understand differences in risk between subgroups of YACs and the diversity in outcomes among YACs [1].

To our knowledge, only a few studies have examined differences in outcomes among YACs depending on type of relationship to the care-receiver. An Australian study compared a mixed-age sample of children, adolescents, and young adults ( age 9–20 years) in healthy families (*n* = 1768) with youth in families with parental illness (*n* = 336) and families with another ill family member in the household (*n* = 116). Youth in families with an ill family member had more mental health problems and lower life satisfaction compared to youth in healthy families. However, youth in families with parental illness had worse adjustment compared to youth in families with another ill family member [11], indicating that the type of relationship between a young carer and the care-receiver may have importance for the adjustment of the carer. Furthermore, youth in families with parental illness reported more caregiving responsibility than youth in families with another ill family member [12].

A study on young adults in Italy (*N* = 1823, age 18–29 years) who cared for an ill/disabled family member during the COVID-19 pandemic found poorer mental health among those who cared for a parent (*n* = 268) compared to those who cared for another ill/disabled family member (*n* = 97), i.e., a sibling or a grandparent [13]. So far, this is the only previous study focusing specifically on YACs where outcomes are examined based on type of relationships with the care-receiver. However, YACs who cared for partners, friends, or someone outside the family were not included. This is unfortunate as emerging adults may well be intimately involved with and potentially also provide care for persons outside of their family of origin. Furthermore, whereas the context of COVID-19 is interesting and important, it probably represents an exceptional situation for YACs, differing in important ways from the challenges involved in caregiving pre- and post-COVID-19, regarding, e.g., income, severity of illness and death, time spent at home, and availability of health care. Thus, we have very little knowledge about how the type of relationship between the YACs and the care-receiver may impact the health and well-being of YACs.

A limited number of studies have examined outcomes among YACs depending on the type of illness in the care-receiver. In the Australian study referenced above [11,12], type of illness in the care-receiver was related to the adjustment of children, adolescents, and young adults (9–20 years). That is, youth in families with physical illness or disability had better adjustment compared to youth in families with either mental illness or substance abuse. However, no differences in the youth’s experience of caregiving were found across type of illness in the care-receiver [12]. Another Australian study (*N* = 81, age 10–25 years) found no differences in outcomes when the parent had mental health problems compared to physical illness/disability [14]. A third study, including both young caregivers and non-caregivers (*N* = 245, age 10–25 years), found a greater adverse impact in youths with parents with mental illness compared to physical illness/disability [15]. Finally, a study on the adjustment of young carers in Norway (*N* = 246, age 8–18 years) found no differences between youths in families with physical illness, mental illness, or substance abuse [6].

To conclude, previous studies have primarily examined parental illness, and no studies have compared a broader range of relationship types between YACs and care-receivers. Additionally, studies on different types of illness in care-receivers have either examined carers younger than 18 years or applied samples with a large age span, comprising children, adolescents, and young adults. Most studies have included relatively small samples, making it hard to identify minor differences between groups. Furthermore, findings have been inconsistent regarding the impact of different illness types on outcomes in young caregivers. To further examine the differences between sub-groups of YACs, studies with larger, representative samples of 18- to 25-year-old carers, preferably including control groups, are warranted.

Studies have reported that on average, YACs spend between 13 and 20 h per week on caring, with the majority having assumed caring responsibilities either before the age of 16 or in the period between 16 and 20 years [1,5,16,17]. However, we lack knowledge on whether the amount of caring provided by YACs is associated with type of illness in the person receiving care and/or type of relationship between the carer and care-receiver. Thus, it remains unknown if caring for a close family member is more demanding regarding daily hours of caring compared to caring for someone outside the family. Additionally, we do not know if caring for someone with mental health problems or substance abuse is more demanding regarding the amount of time required compared to someone with a physical illness or disability. Quantitative studies have to a large degree looked at YACs as a homogeneous group across context variables. This has been a reasonable strategy in view of small sample sizes and limited knowledge about experiences, mental health, and well-being among YACs. To further increase our understanding of YACs, it is reasonable to look at context variables that may differentiate between sub-groups of YACs.

In the present study, we examine differences in mental health and life satisfaction among YACs by comparing the type of relationship between YACs and the care-receiver. We also explore whether caring for a close relative requires more time compared to caring for a more distant relative, a partner, or someone outside the family. Furthermore, we examine differences in mental health and life satisfaction in YACs associated with type of illness in the care-receiver. The aim is to contribute to a more comprehensive, in-depth understanding of how informal care may impact the health and well-being of YACs.

The following research questions will be examined:(1)Do young adult carers have more mental health problems and lower life satisfaction compared to young adult students without care responsibilities?We hypothesize that YACs will report more mental health problems and lower life satisfaction.(2)Does type of relationship between YAC and the care-receiver have different impact on the mental health and life satisfaction of YACs? The relationships investigated are the care-receiver being a close relative (siblings, parent or grandparent), a partner, someone outside the family (friend, neighbor, etc.), or another relative (aunt, uncle, cousin, etc.).Due to limited evidence from previous studies, we have not made hypotheses regarding which type of relationship expected to t be strongest associated with negative outcomes in YACs.(3)Does type of illness in the care-receiver have different impact on the mental health and life satisfaction of YACs? The illnesses examined are physical illness, mental health problems/disorders, impaired functioning, and drug or alcohol abuse.Based on previous findings, we hypothesize that YACs caring for someone with mental illness or alcohol and/or drug abuse will have poorer adjustment compared to YACs caring for someone with physical illness or disability.(4)Is the amount of care provided by YACs associated with type of relationship between YACs and care-receiver or type of illness in the care-receiver?To our knowledge this question has not been examined previously. Thus, we have no specific hypotheses regarding what type of illness or relationship may demand most hours of daily care from YACs.

## 2. Materials and Methods

### 2.1. Procedure

The SHoT study (Students’ Health and Wellbeing Study) is a large survey of students in Norwegian higher education conducted by the three largest student welfare organizations. Four surveys have been completed since 2010. This report is based on the most recent wave, conducted in 2022. Detailed information of the SHoT study has been described in a previous publication [18]. Although some universities and colleges allocated time in school classes allowing the student to complete the survey during a lecture, no teachers were instructed to provide support or assistance. Students were told that participation was completely voluntary, and that there were no penalties for not filling out the survey. The average time spent answering the questionnaire was 30 min.

### 2.2. Participants

Data from SHoT2022 were collected from February to April 2022 and included full-time students undertaking higher education in Norway. All 169,572 students enrolled in higher education in Norway received an invitation to participate, of whom 59,554 students completed the web-based questionnaires (response rate: 35.1%). As the current study was an investigation of YACs, we excluded participants aged 26 years and older, yielding a final sample size of 37,731 participants, aged 18–25 years.

### 2.3. Ethics

The SHoT2022 study was approved by the Regional Committee for Medical and Health Research Ethics in Western Norway (no. 2022/326437). Informed consent was obtained electronically after the participants had received detailed information about the study.

### 2.4. Instruments

#### 2.4.1. Exposure Variable

All students were asked if they had regular care responsibilities for someone with physical or mental illness, disabilities, or drug or alcohol abuse. If they answered yes to this question, they were asked about the number of hours they spent on a typical weekday to help this person (or persons). Details and exact phrasing of questions are found in Table 1. The questions have previously been tested for clarity among young carers (5–17 years) and their parents [19].

#### 2.4.2. Sociodemographic and Lifestyle Information

Data about the participants’ age and gender were extracted from their 11-digit Norwegian national identity number, and all participants were asked about their relationship status. In terms of country of origin, participants were categorized as an immigrant if either the student or their parents were born outside Norway.

#### 2.4.3. Mental Health Problems

Mental health problems were assessed using the Hopkins Symptoms Checklist (HSCL-25) [20], derived from the 90-item Symptom Checklist (SCL-90), which is a screening tool designed to detect symptoms of anxiety and depression. It is composed of a 10-item subscale for anxiety and a 15-item subscale for depression, with each item scored on a 4-point scale ranging from “not at all” (1) to “extremely” (4). The period of reference is the prior two weeks. An investigation into the factor structure based on the SHoT2014 dataset showed that a unidimensional model, in contrast to the original subscales of anxiety and depression, has optimal psychometric properties for application to student populations [21]. In the current study, the HSCL-25 was analyzed as a continuous average score of all 25 items (range 1–4). The Cronbach’s alpha of the HSCL-25 in the current study sample was 0.94.

#### 2.4.4. Life Satisfaction

Life satisfaction was assessed by the Satisfaction with Life Scale (SWLS) [22]. The SWLS is a 5-item scale designed to measure global cognitive judgments of one’s life satisfaction and is not a measure of either positive or negative feelings. Participants indicate how much they agree or disagree with each of the 5 items using a 7-point scale ranging from “strongly agree” (7) to “strongly disagree” (1). In the current study, the SWLS was analyzed as a continuous total score (range 5–35). The Cronbach’s alpha of the SWLS in the current study sample was 0.89.

### 2.5. Statistics

IBM SPSS version 28 (SPSS Inc., Chicago, IL, USA) for Windows was used for all analyses. Group differences between causes of care and relationship were examined using the General Linear Model (GLM) with Least Significant Difference (LSD) post hoc multiple comparisons, calculating estimated marginal means (EMM) for the two outcome measures (HSCL-25 and SWLS), controlling for age and sex. There was generally very little missing data (*n* < 0.5%); hence, missing values were handled using listwise deletion. As the SHoT2022 study had several objectives and was not designed to specifically study circadian preference, no a priori power calculations were conducted to ensure that the sample size had sufficient statistical power to detect differences in outcomes. Still, it should be noted that the sample comprised 38,000 respondents.

## 3. Results

### 3.1. Sample Characteristics

The total sample comprised 37,731 young adults (68% women), with a mean age of 22.3 years (SD: 1.5). As detailed in Table 2, the majority of the participants were single (55.6%), and approximately 10% were considered immigrants to Norway.

### 3.2. Care Responsibilities

A total of 5.4% (*n* = 2017) of the total sample reported having care responsibilities for others, and the proportion was higher among female (6.1%, *n* = 1562) than male (3.8%, *n* = 455) students. As detailed in Table 3, the most common relationship with the care-receiver was “a close relative” (3.4% of the whole sample), followed by “someone outside the family” (1.8%), “a partner” (0.5%), and “another relative” (0.3%).

As detailed in Figure 1, the most common type of illness in the care-receiver was mental health problems or disorders (46.7%), followed by drug or alcohol use (24.8%), impaired functioning (19.7%) and physical illness (8.8%). This pattern was similar for male and female students.

### 3.3. Mental Health among YACs

Figure 2 shows the level of mental health problems and satisfaction with life stratified by relationship with the care-receiver and type of illness in the person receiving care. As shown in Figure 2A caring for a partner was associated with the highest level of mental health problems in YACs. Additionally, while there were only minor differences between the other relationship categories, all categories were significantly associated with more mental health problems compared with students without care responsibilities (all *p* values < 0.001). Figure 2C shows that there were no significant differences between relationship categories in terms of life satisfaction, although all categories here were also significantly associated with poorer life satisfaction compared with students without care responsibilities (all *ps* < 0.001).

As shown in Figure 2B, caring for a person affected by drug or alcohol abuse was associated with the highest level of mental health problems among the carer, whereas caring for a person with a physical illness was associated with the lowest level of mental health problems. However, all four groups of carers had significantly worse mental health compared to students without care responsibilities (all *ps* < 0.001). The opposite pattern was observed when investigating life satisfaction as the outcome variable. As detailed in Figure 2D, those caring for a person with drug or alcohol abuse had the lowest life satisfaction compared to all other carers. No other significant group differences were observed, although all four groups of carers scored significantly lower compared with students without care responsibilities (all *ps* < 0.001). The patterns were similar for both men and women, and there were no significant gender interactions for any of the analyses.

### 3.4. Hours of Daily Care

YACs reported spending on average 2:12 h per day on caring. In terms of the relationship with the care-receiver, caring for a partner was associated with significantly more hours of daily care (2:55 h per day), compared with 2:03–2:13 h per day for the other categories (see Figure 3A for details).

As detailed in Figure 3B, the number of daily hours spent on care was relatively similar across all type of illness in the care-receiver, although mental disorders were associated with less care (2:01 h per day), compared with impaired functioning (2:25 h per day).

There were no significant gender interactions for any of the analysis.

## 4. Discussion

The aim of the present study was to provide knowledge on the relationship between the caregiving context and outcomes among YACs. We examined data from a national survey of college and university students in Norway. Four issues were examined: (1) mental health and life satisfaction in YACs compared to non-caring young adults, (2) the impact of type of relationship between YACs and the care-receiver on YACs’ mental health and life satisfaction, (3) the impact of type of illness in the care-receiver on the mental health and life satisfaction of YACs, and (4) differences in amount of care provided in different caregiver contexts.

More mental health problems and lower life satisfaction were found among YACs compared to students without care responsibilities. YACs caring for a partner had the highest level of mental health problems, followed by YACs caring for a close relative. Furthermore, caring for a partner was associated with significantly more hours of daily care compared to all other types of relationships. YACs providing care for a person with substance abuse reported higher levels of mental health problems and lower levels of life satisfaction compared to all other illness categories, followed by mental illness and physical illness/disability. The number of hours spent on daily care was relatively similar across all illnesses, except for less amount of time spent caring when the care-receiver had mental health problems compared to caring for someone with impaired functioning.

### 4.1. YACs Outcome Compared to Control Group

YACs in the present study reported more mental health problems and lower life satisfaction compared to students without care responsibilities. Thus, previous findings of negative outcomes in YACs were confirmed [1,2,3,4]. Poorer outcomes were reported across the type of relationship between YACs and care-receiver and type of illness in the person receiving care. Therefore, having care responsibility was associated with more mental health problems and lower life satisfaction regardless of whether the care-receiver was a close relative (e.g., sibling or parent), someone outside the family (e.g., aunt or cousin), a partner, or another relative (e.g., friend or neighbor). Additionally, outcomes were poorer for YACs compared to students without care responsibilities regardless of whether the care-receiver had mental health problems/disorders, substance abuse, physical illness, or disability. This points to the need to attend to the situation of YACs regardless of who the care-receiver is or what type of illness they are suffering from.

### 4.2. Type of Care Relationship and Hours of Care

The most frequent care relationship was to “a close family member” including siblings, parents, or grandparents. More surprising was the number of students reporting providing care for someone outside the family, such as a friend or neighbor. This type of relationship has been given little attention in previous studies on YACs. It is a natural part of the developmental stage of emerging adults to establish close relationships outside the family. Furthermore, these relationships may also include caregiving, as suggested by our results. Many of the students (almost 45%) reported having a partner. Only a small number of these reported providing care for their partner, but these had the highest level of mental health problems and spent the highest number of hours caring on a typical weekday.

Although partners are included in the commonly referred-to definition of YACs by Becker and Becker [5], to our knowledge, no previous study on YACs has examined partners or other care-receivers outside of the family. This may partly be due to small sample sizes where only the most frequently occurring caring relationships have been included. It may also be influenced by studies on young carers (<18 years) where being involved in a steady relationship with a partner or caring for someone outside the family is not expected.These relationships have perhaps not been studied because one simply has not recognized the importance of care relationships outside the family.

YACs spent on average 2:12 h each day on caring activities. Caring for a partner, however, was associated with significantly more hours of daily care (2:49 h per day), compared to all other types of relationships. A dose–response pattern between number of hours of daily caring and negative health outcomes, as well as poorer study progress and loneliness, has previously been reported among YACs [7,9]. Thus, the increased hours of daily care when caring for a partner may to some degree explain the higher level of mental health problems observed in this sub-group of YACs.

Another explanation of the negative outcomes in those caring for a partner may be the similarity in mental health problems often found between partners at the start of a relationship [23]. Thus, higher level of mental health problems in YACs caring for a partner may be a result of assortative mating, with mental health problems in the YAC being present prior to establishing a caring relationship with a partner.

The higher level of mental health problems in this group of YACs may also be explained by the strain of caring for a partner who you spend most of your hours with outside school or work. It has been argued that providing care for a partner may be more burdensome for a young carer compared to someone older because of the many roles and responsibilities needed to be balanced in this phase of life, e.g., education, career, parents, and children [24].

Furthermore, it is possible that YACs who enter a care relationship with a partner may have learned to take the role as caregiver through previous or ongoing care for close family member(s). We could speculate about whether this has made them expect non-reciprocal relationships with others, characterized by an uneven balance between receiving and providing care. Thus, as young adults, they might continue to take on care responsibilities in new relationships, and perhaps even end up caring for several persons, e.g., both a close family member and a partner. A study among Dutch students with caring responsibility found that 19% were caring for more than one person [16]. We would like to see further studies on the development of care relationships between YACs and their partners, as well characteristics of YACs who have care responsibility for multiple persons.

Apart from those caring for a partner, the group of YACs with the highest levels of mental health problems were those caring for a close relative. This group of YACs had poorer mental health than those caring for someone outside the family or for another relative. Previous studies on young children, adolescents and young adults have reported more mental health problems when caring for an ill parent [11,13]. Our findings are difficult to compare to these studies as they examine caring for a parent versus for another close family member, without including YACs caring for a more distant relative, a partner, or someone outside the family.

No differences were found between type of care relationship and life satisfaction. Regardless of who the YACs provided care for, they reported lower life satisfaction compared to students without care responsibilities. Thus, when including positive outcome measures, poorer outcomes are also demonstrated in YACs.

We do not have any data to examine the various explanations discussed above. Combined with the limited number of previous studies on different types of care relationships, the mechanisms regarding associations between care relationship and mental health problems in YACs remains unsubstantiated.

### 4.3. Type of Illness in Care-Receiver and Hours of Care

The most common type of illness in the care-receiver was mental health problems or disorders, followed by drug or alcohol abuse (termed substance abuse hereafter), with almost half of the YACs caring for someone with mental health problems/disorders and about 25% caring for someone affected by substance abuse. Caring for a person affected by substance abuse was associated with the highest level of mental health problems and lowest level of life satisfaction in YACs compared to all other illness groups. Furthermore, caring for a person with mental health problems was associated with poorer outcomes in YACs compared to physical illness or disability. Besides substance abuse, no differences between illness groups were observed regarding life satisfaction.

Previous studies have reported associations between mental illness in care-receivers and negative outcomes in children, adolescents, and young adult carers [11,15]. However, we have not found any studies specifically reporting associations between negative outcomes in YACs and caring for someone affected by substance abuse. When studies have included substance abuse as a separate illness category, this has comprised a very small proportion of the samples, e.g., [6,11]. In addition to age differences compared to the present study, this may explain why a higher level of mental health problems in YACs caring for someone with substance abuse has not previously been observed.

The number of daily hours spent on caregiving was relatively similar across all illnesses, except for YACs caring for someone with mental disorders, who reported lower hours of daily care compared to caring for someone with impaired functioning. The illness category requiring most hours of daily caring was impaired functioning, followed by physical illness. Thus, the number of hours of daily care cannot explain the higher level of mental health problems in YACs caring for persons with substance abuse and/or mental illness. Hence, other factors may be stronger predictors of YAC outcomes, for example illness characteristics, type of care tasks provided, and/or level of emotional burden and relational stress in the caregivers.

The increased mental health problems and low life satisfaction reported by YACs providing care for persons affected by substance abuse may be understood in view of the dysfunctional interaction patterns often associated with substance abuse. Whereas positive and rewarding relationships may exist, many studies have demonstrated increased risk for various dysfunctional interaction patterns related to parental substance abuse, e.g., unpredictability [25], increased family conflict [26], lower family cohesion, and a more negative emotional climate [27], as well as increased risk of violence, neglect, and abuse [28,29]. These are interaction patterns that may heighten the level of caregiver strain, probably regardless of the type of relationship between YACs and the care-receiver [30]. Furthermore, comorbid mental health disorders are common in people affected by substance abuse [31] and may represent additional challenges in the care relationship. For some, caring for a person who abuses substances may involve coping with anger outbursts, intoxicated behavior, and/or fear of overdose/delirium, accidents, or sudden death. Thus, caring for someone affected by substance abuse may be more emotionally complex and draining, partly explaining the increased mental health problems and lower life satisfaction among YACs in this illness category, compared to those caring for persons with physical or mental illness, or disability.

Both substance abuse and mental illness are characterized by stigmatization and shame [32,33], increasing the likelihood that both the illness and the caregiving situation remain hidden. Fear of stigma may represent a barrier for carers to seek help, leaving YACs with less acknowledgement and limited support. Accordingly, a study on young carers found that parents with physical illness received more social support from the network and more formal care (e.g., practical home-based services) compared to parents with mental illness or substance abuse [6]. If this is a general pattern, family members with substance abuse and/or mental illness are more dependent on informal care from family members, including YACs, compared to those with physical illness. Additionally, more caregiving discomfort has been reported by children, adolescents and young adults caring for a parent with mental illness compared to physical illness/disability (while substance abuse was not explicitly identified). These findings have been interpreted as a product of the stigma associated with mental illness [14].

Type of care tasks may to some degree differ between illness categories and partly explain the differences in outcomes between YACs caring for someone with substance abuse or mental illness versus physical illness. Care-receivers suffering from substance abuse and/or mental illness are assumed to require more emotional care compared to those with physical illness/disability [34,35,36]. The literature on parent–child role-reversal/parentification argues that emotional care tasks, e.g., proving support, comfort, discipline, and entertainment, diverting negative thoughts, or providing supervision, are more complex and burdensome, and also more harmful for young carers than practical tasks [16]. Emotional care tasks are often characterized by vague expectations, requiring a high level of psychological maturity and emotion regulation in the carer, and may provide the carer without much acknowledgement for their contribution [37]. The negative impact of emotional care tasks has received some support in a study of young carers [38]. In accordance with this, YACs describe a high level of emotional care to be particularly difficult, being tasks that restrict their participation in other life events and that take time away from rest, relaxation, and socializing [5]. However, findings regarding the impact of different types of care tasks have so far been inconsistent. Personal care tasks, as well as financial and practical tasks, have been found to predict more negative outcomes among young carers younger than 18 years, whereas emotional care tasks were not associated with youth outcomes [6]. Another study including a mixed-age sample (11–24 years) found participation in social and emotional care tasks to be associated with better adjustment, i.e., higher quality of life and less externalizing behavior in the carers [39]. However, the type of illness included in this study was predominantly physical illness, with only 6.7% of the care-receivers categorized as suffering from mental illness and 12.5% from substance use. We expect the relationship between providing emotional care tasks and mental health in YACs to be complex, and perhaps also to differ according to the dimension of mental health outcome being assessed (e.g., externalizing versus internalizing symptoms). This complexity is illustrated in a recent study demonstrating interrelated patterns between type of illness in the care-receiver, type of care tasks, and mental health of YACs [36]. The inconsistent findings indicate the need for studies to determine whether different types of care tasks are related to different illness categories and if this may explain the differences in outcomes among YACs. Future studies need to focus specifically on YACs and include substance abuse among the illness categories examined.

Additionally, other characteristics associated with substance abuse and/or mental illness may be relevant to explain the results in the present study. A study on youth and young adults (10–25 years) found that parental mental illness had a more gradual onset compared to physical illness/disability. Furthermore, the association between gradual onset of parental illness/disability and poorer adjustment in the youth was demonstrated [14]. Thus, a more gradual onset and a somewhat more complex etiology in substance abuse and mental illnesses compared to some types of physical illness/disability may make these illnesses more difficult to understand and cope with, perhaps contributing to poorer outcomes in the young caregivers.

Finally, genetic components have been demonstrated in the etiology of substance abuse as well as a range of mental illnesses [40]. Thus, among YACs who provide care for a close family member affected by substance abuse and/or mental illness, we cannot exclude the possibility of common genes contributing to associations between care-receiver illness and increased mental health problems in the caregivers.

We need to consider multiple factors to explain the increased levels of mental health problems and reduced life quality in YACs caring for someone with substance abuse and/or mental health problems/disorders. Further studies are needed where more information is included on characteristics related to type of illness/disability in the care-receiver and the experiences of caring among YACs (e.g., functional impairment and severity of illness, illness duration, gradual vs. sudden onset of illness, level of stigma, experiences of predictability, controllability, and coping in the YAC, type of caring tasks provided, etc.). Hopefully, this will shed light on possible mechanisms explaining the differences in outcomes among YACs related to type of relationship to and type of illness in the person receiving care.

### 4.4. Strengths and Limitations

The present study comprises a large sample of YACs, examining questions that have been given little attention in previous studies, i.e., the impact of type of relationship between carer and the person receiving care and type of illness in the care-receiver. The sample was derived from a national study of all Norwegian students pursuing higher education. The large sample size allows us to differentiate between subgroups of YACs and to identify associations not previously found or examined due to small samples and/or samples predominantly including care-receivers with physical illnesses. The present sample also enables us to compare outcomes between young adults with and without care responsibilities.

Another strength of the present study is the use of psychometrically sound outcome measures, including assessment of both psychological distress and well-being among YACs. By assessing a broader range of relationships, both inside and outside the family context, new knowledge has emerged, demonstrating the importance of investigating young adults as caregivers not only for close family members, but also for persons outside the family of origin, e.g., partners. As we did not distinguish between caring for parents versus other family members, we were not able to replicate findings from previous studies demonstrating differences between these roles.

The cross-sectional method applied represents a limitation. Thus, we cannot determine the temporal order or causality between type of relationship or type of illness in the care-receiver and mental health problems or life satisfaction in YACs. However, type of relationship and type of illness in the caregiver most likely affects mental health and life satisfaction in the caregiver, rather than the other way around (except if assortative mating is a factor that contributes to the association between caring for a partner and negative outcomes in YACs). Another limitation in the present study is the selection bias present in a sample of young adults in higher education, where young adult carers who are in training, employment, or on welfare benefits are not included. This is a limitation considering that YACs may experience barriers against entering higher education [41]. Thus, further studies should also include YACs outside of higher education. As women and girls constitute about 70% of the student population in Norwegian colleges/universities, the sex difference present in our sample should not represent a substantial bias. However, the results should be interpreted in accordance with the relatively modest response rate for the present survey (35.1%).

The YACs in the present study were identified through self-report. Young carers are often not identified by professionals in health care, education, and social services [42], and as far as we know, no organizations or services offer support to YACs in Norway. Therefore, self-report was considered the best available strategy to identify this group of carers. The survey questions applied to identify YACs have previously been thoroughly examined for clarity and for the suitability for identifying young carers [19]. Furthermore, the percentage of students who reported care responsibilities in the present study was almost identical to the prevalence found in a previous wave of the survey, i.e., 5.5% in the 2018 wave versus 5.4% in the present study [7]. Additionally, the proportion of female versus male YACs was similar, i.e., 6.4% female and 3.4% male in 2018 versus 6.1% female and 3.8% male in the present study. This consistency strengthens our confidence in the procedure used to identify YACs.

Cultural differences may exist regarding the degree to which young people are expected to provide informal care and how much their caregiving is valued and acknowledged by others. As the Norwegian healthcare system is expected to provide basic healthcare for all citizens, the needs and experiences of YACs are rarely recognized. Thus, the findings in the present study may be less generalizable to non-Western countries and countries without a strong welfare state, where providing informal care may be more common and more culturally expected, as well as YACs receiving more credit for contributing as caregivers.

Finally, a limitation in the present study is the lack of information about psychological and social mechanisms that could explain the poorer outcomes among the YACs providing care for partners and/or caring for someone affected by substance abuse. We anticipate further studies that may increase our understanding of these associations.

### 4.5. Practical Implications

Authorities as well as professionals within healthcare, social services, and the educational system should be sensitized to the needs of YACs. The high-risk status of young adults who provide care for someone affected by substance abuse and/or mental illnesses in a partner or a close family member needs to be acknowledged, and increased support, information, and practical help must be made available. Home-based services to care for mentally and physically ill patients as well as those who abuse substances should be offered. When necessary, referrals should be made to specialized services for the young carers themselves. Finally, heightened public awareness and appreciation for the contributions made by YACs could increase their positive experiences of providing care. Research on YACs needs to continue, with emphasis on identifying sub-groups of high-risk individuals. Along with providing more support for YACs, we need to evaluate the outcomes of different types of services.

## 5. Conclusions

The present study is based on a representative sample of Norwegian students in higher education, with 5.4% of the students reporting care responsibility. Students not reporting care responsibilities were included as a control group. Hence, the study fulfils the methodological requests raised in the YAC literature about more representative samples and inclusion of a control group [4]. The majority of the YACs in this study provided care for a close relative, with the largest group of care-receivers being persons with mental health problems/disorders. The YACs had more mental health problems and lower life satisfaction compared to students without care responsibilities. Most previous studies on YACs include only close family members as care-receivers. By expanding the type of relationships to also include partners and persons outside the family of origin, new knowledge emerged. Two groups of YACs reported poorer outcomes. compared to other students with care responsibilities. These were YACs who provided care for a partner and YACs who provided care for someone with substance abuse. These are subgroups of YACs that need to be acknowledged, given further attention, and offered support.

YACs are expected to rise in numbers in the years to come, due to increased reliance on home-based care provided by family members, a growing elderly population, and a growing number of persons living with chronic illnesses due to advances in medical treatment [43]. Given the results in the present study and knowledge from the growing research literature on YACs, it is worrisome that the needs of these informal carers are unrecognized, with minimal or no support services attending to their needs, neither within health services nor in the educational system.

The findings in the present study raise questions that need to be examined further, such as how to explain the differences in YAC outcomes related to different types of relationships and various illness categories. Thus, further studies related to the care context of YACs are warranted, where mediating variables are included that can explain associatins between types of caregiver/care-receiver relationship, illness categories, and outcomes in YACs.

## Figures and Tables

**Figure 1 ijerph-20-03925-f001:**
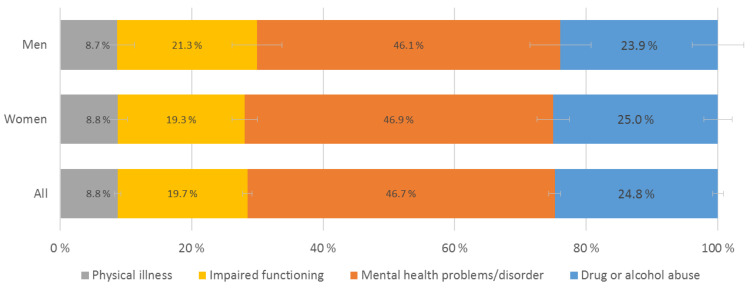
Distribution of type of illness in care-receiver.

**Figure 2 ijerph-20-03925-f002:**
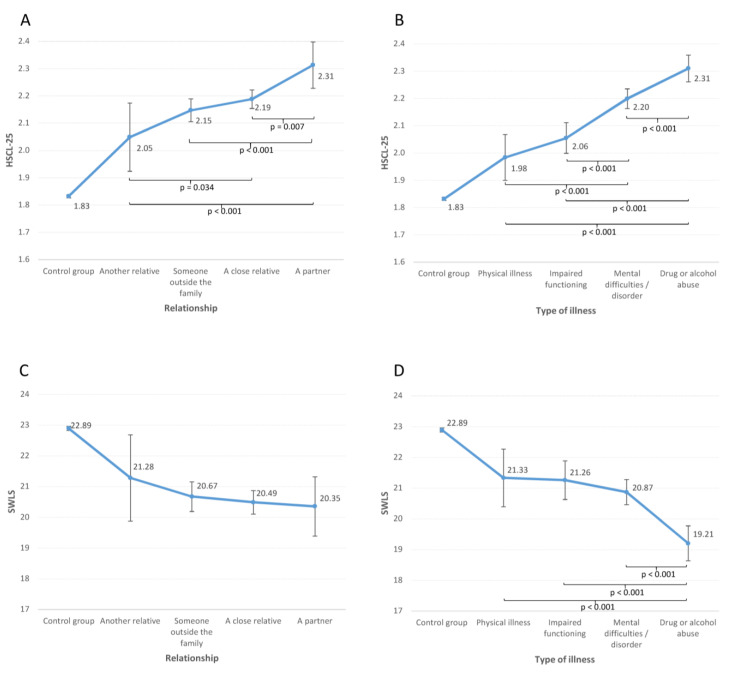
Mental health problems (HSCL-25) and life satisfaction (SWLS) by type of relationship between YACs and care-receiver (**A**,**C**) and type of illness in care-receiver (**B**,**D**).

**Figure 3 ijerph-20-03925-f003:**
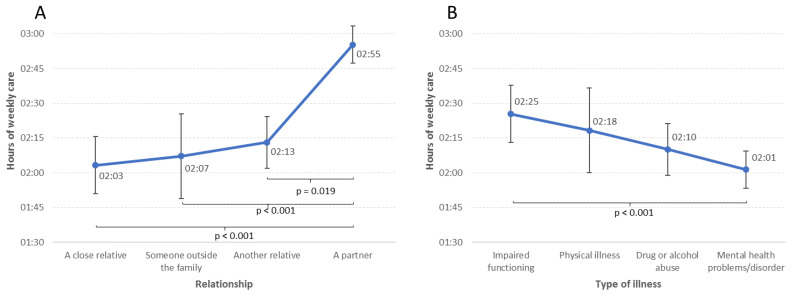
Hours of daily care by type of relationship between YAC and care-receiver (**A**) and type of illness in care-receiver (**B**).

**Table 1 ijerph-20-03925-t001:** Questionnaire assessing care responsibilities.

**Other care responsibilities**
Some people provide help or support to people who are physically or mentally ill, have a disability, or who abuse alcohol or drugs. This can be parents, siblings, other relatives, or others.
**Do you have any such people that you need to look after regularly?**
◯ Yes ◯ No
If Yes:
**Your relationship with this person (s):**
*Check one or more boxes.*
☐ A partner
☐ A close relative (siblings, parents or grandparents)
☐ Another relative (aunt, uncle, cousin, etc.)
☐ Someone outside the family (friend, neighbor, etc.)
**What problem (s) does the person receiving your help and support have?**
*Check one or more boxes.*
☐ Physical illness
☐ Mental health problems/disorder
☐ Impaired functioning
☐ Drug or alcohol abuse
**Approximately how many hours do you spend on a typical weekday to help this/these person (s)?** Drop-down menu: 0–18 h. or more

Note: When recoding the 4 relationship variables (responses were not mutually exclusive) into one ordinal variable, a positive response for the “closest” relationship (partner) had precedence.

**Table 2 ijerph-20-03925-t002:** Sociodemographic characteristics of the study sample.

Participant Characteristics	Total (*n* = 37,731)
Age, mean (SD)	22.3 (1.5)
Gender	
Women, % (*n*)	68.0 (25,663)
Men, % (*n*)	32.0 (12,068)
Single, % (*n*)	55.6 (20,981)
Country of origin, % (*n*)	
Norway	90.3 (34,069)
Immigrants	9.7 (3662)

**Table 3 ijerph-20-03925-t003:** Prevalence of type of care relationship in male and female college and university students.

	All *n*		Women *n*		Men *n*	
A close relative	3.40%	(1278)	4.10%	(1053)	1.90%	(225)
Another relative	0.30%	(101)	0.40%	(91)	0.10%	(10)
A partner	0.50%	(204)	0.50%	(**128**)	0.60%	(76)
Someone outside the family	1.80%	(677)	2.00%	(504)	1.40%	(173)
Any *	5.40%	(2017)	6.10%	(1562)	3.80%	(455)

* Any of the 4 relationship types.

## Data Availability

The datasets for this article are not publicly available because of privacy regulations from the Norwegian Regional Committees for Medical and Health Research Ethics (REC). Requests to access the datasets should be directed to BS (borge.sivertsen@fhi.no). Guidelines for access to SHoT data are found at https://www.fhi.no/en/more/access-to-data (accessed on 19 January 2023). Approval from REC (https://helseforskning.etikkom.no, accessed on 19 January 2023) is a pre-requirement.

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
