# Peer review of "Does It Matter Who You Provide Care for? Mental Health and Life Satisfaction in Young Adult Carers Associated with Type of Relationship and Illness Category—A National Student Survey"

_ijerph, 2023, doi:10.3390/ijerph20053925_

Round 1

Reviewer 1 Report

Congratulations to the authors for the study. The subject matter is relevant and the presentation of the manuscript is generally good. Here are some suggestions:

-In sub-section of present study: the hypotheses should be presented more clearly, (H1, H2...). Are the questions different from the hypotheses? This question may cause confusion.

-More detailed procedure.

-Include a sub-section on participants before or after the procedure sub-section (although the results of the sample descriptions should be kept in the results section). 

-Table from the responsibility assessment questionnaire to be included as an annex.

-In the statistics section, provide details of the type of analysis to be carried out. 

-I think the question of "non-Norwegian" needs to be clarified, I am confused, only Roma participants were considered "non-Norwegian", people born in Norway but with foreign parents were considered "non-Norwegian", this question should be better explained and I would opt for another nomenclature for the category, rightly omit it.

Furthermore, the distinction between the two categories is not understood when no question of provenance is included in the hypotheses, also emphasising the poor statistical representation as a salient fact. 

-In the results section, in addition to the graphs with the relationships, I think it would be clearer to add tables with the analyses and relationships. 

-In the discussion section I would add a sub-section on "practical implications". 

-In the limitations it would also be interesting to refer to the gender perspective in care. In this section it could be interesting to highlight the importance of considering cultural aspects or aspects of origin for future studies. 

Author Response

We thank the reviewer for the time and effort invested in studying and suggesting revisions to our manuscript. We have revised our manuscript as thoroughly as possible according to the comments received.

REVIEWER 1:

Comments and Suggestions for Authors

Congratulations to the authors for the study. The subject matter is relevant and the presentation of the manuscript is generally good. Here are some suggestions:

  • In sub-section of present study: the hypotheses should be presented more clearly, (H1, H2...). Are the questions different from the hypotheses? This question may cause confusion.

    Response: We agree that this could be made clearer. The presentation of research questions and hypotheses have now been revised. Hopefully they are now clearer and have a better correspondence between research questions and hypotheses. Page 4 and 5, line 190 to 219.

  • More detailed procedure. Include a sub-section on participants before or after the procedure sub-section (although the results of the sample descriptions should be kept in the results section). 

    Response: We have provided more details in the Procedure section and have also added a new sub-section on participants, as suggested by the reviewer. Page 5 and 6, line 254 to 277.
  • Table from the responsibility assessment questionnaire to be included as an annex.

    Response: We prefer to keep the responsibility questionnaire (Table 1) in the main body of the manuscript and not move it to an appendix. The reason for this is that we want to maintain as much transparency as possible about how YACs were identified in this study, as well as making it easier for the reader to evaluate the findings. We will move Table 1 to appendix if editor consider that this will improve the manuscript. Page 6 and 7, line 295 to 297.

  • In the statistics section, provide details of the type of analysis to be carried out. 

    Response: We have now provided more details on the statistical analyses:

    “Group differences between causes of care and relationship were examined using the General Linear Model (GLM) with Least Significant Difference (LSD) post hoc multiple comparisons, calculating estimated marginal means (EMM) for the two outcomes measures (HSCL-25 and SWLS), controlling for age and sex.” Page 8, line 335 to 336.

  • I think the question of "non-Norwegian" needs to be clarified, I am confused, only Roma participants were considered "non-Norwegian", people born in Norway but with foreign parents were considered "non-Norwegian", this question should be better explained and I would opt for another nomenclature for the category, rightly omit it.

    Response: We agree with this comment, and we have now rephrased this sentence in the Methods section, and omitting the term “non-Norwegian”:

    “In terms of country of origin, participants were categorized as an immigrant if either the student or his/her parents were born outside Norway.” Page 7, line 301 to 303.
  •  
  • Furthermore, the distinction between the two categories is not understood when no question of provenance is included in the hypotheses, also emphasising the poor statistical representation as a salient fact. 

  • Response: We agree that the distinction between “immigrants” versus “Norway being country of origin” is not a part of the research question or hypotheses. Furthermore, this variable is not controlled for in any analyses and might therefore be taken out of the manuscript. However, we have kept the distinction so far as this is a distinction to describe the sample instead of ethnicity that makes less sense in a Norwegian context. If the editor wishes, we have no problems with removing it altogether for the manuscript. (Page 8, line 350 and 351 + Table 2) 

  • In the results section, in addition to the graphs with the relationships, I think it would be clearer to add tables with the analyses and relationships. 

    Response: While we agree with the reviewer that tables are useful to provide a detailed representation of the statistical results, we are not sure if the readability of the paper will improve if we present both figures and details outlining nearly identical results. As such, we have opted to not add further tables to the manuscript.

In the discussion section I would add a sub-section on "practical implications". -In the limitations it would also be interesting to refer to the gender perspective in care. In this section it could be interesting to highlight the importance of considering cultural aspects or aspects of origin for future studies. 

Response: A section on “Practical implications” has now been added to the discussion

 (Page 19)

Authorities as well as professionals within health care, social services, and the educational system should be sensitized to the needs of YACs. The high-risk status of young adults who provide care for someone with substance abuse and/or mental illnesses in a partner or a close family member needs to be acknowledged, and increased support, information, and practical help made available. Home-based services to care for mentally and physically ill as well as substance abusers should be offered. When necessary, referrals should be made to specialized services for the young carer him/herself. Finally, heightened public awareness and appreciation for the contributions made by YACs could increase their positive experiences of providing care. Research on YACs need to continue, with emphasize on identifying sub-groups of high-risk individuals. Along with providing more support for YACs, we need to evaluate the outcomes of the different types of services.”

In the limitation section the following comments about cultural aspects have been included: (Page 19)

“Cultural differences may exist regarding the degree young people are expected to provide informal care and how much their caregiving is valued and acknowledged by others. As the Norwegian health care system is expected to provide basic health care for all citizens, the needs and experiences of YACs are rarely recognized. Thus, the findings in the present study may be less generalizable to non-Western countries and countries without a strong well-fare state, where providing informal care may be more common, more culturally expected, as well as YACs receiving more credit for contributing as caregivers”.

Reviewer 2 Report

This is a ecological study on the quality of life of young persons referring taking care of someone with healths or mental problems.  It is based on a national Norway study of quality of life of universitary students. The design, problem and conclusion of the manuscript were adequately presented and conclusions are sound. 

The only problem is the option for univariate analysis without interaction on variables analyzed. The data appears so good and large that a logistic regression of quality of life must show some relationships with the other variables, for example: it is important that partner care is affecting QOL, but this is related to gender, marriage or other variable?

The suggestion of including those analysis will improve the conclusions.  

Author Response

We thank the reviewer for the time and effort invested in studying and suggesting revisions to our manuscript. We have revised our manuscript as thoroughly as possible according to the comments received.

 REVIEWER 2:

Comments and Suggestions for Authors

This is an ecological study on the quality of life of young persons referring taking care of someone with health or mental problems.  It is based on a national Norway study of quality of life of university students. The design, problem and conclusion of the manuscript were adequately presented and conclusions are sound. 

  • The only problem is the option for univariate analysis without interaction on variables analysed. The data appears so good and large that a logistic regression of quality of life must show some relationships with the other variables, for example: it is important that partner care is affecting QOL, but this is related to gender, marriage or other variable?

    Response: This is an important point, and we fully agree that potential moderators and mediators are of great interest, and should be further explored in detail. In the current paper, however, we have chosen to statistically adjust for gender (and age), rather than stratifying the figures by gender. While stratification would be useful if the associations differed between male and female students, this wasn’t the case in the current study – all gender x YAC interaction analyses were non-significant. This information has now been added to the results section (see page 9 line 398 to 400 and page 11 line 419). In terms of marriage, the current sample was quite young (avg. 22 yrs.) and comprised students only.  As such, very few reported to be married, making it difficult to include this variable in the analyses of YACs, despite the large overall sample size.